# CHARTING AND NAVIGATING THE SPACE OF SOLUTIONS FOR RECURRENT NEURAL NETWORKS

**Elia Turner**
Department of Mathematics
Technion, Israel Institute of Technology
`eliaturner@campus.technion.ac.il`

**Kabir Dabholkar**
Department of Mathematics
Technion, Israel Institute of Technology
`kabir@campus.technion.ac.il`

**Omri Barak**
Rappaport Faculty of Medicine and Network Biology Research Laboratory
Technion, Israel Institute of Technology
`omri.barak@gmail.com`

## Abstract

In recent years Recurrent Neural Networks (RNNs) were successfully used to model the way neural activity drives task-related behavior in animals, operating under the implicit assumption that the obtained solutions are universal. Observations in both neuroscience and machine learning challenge this assumption. Animals can approach a given task with a variety of strategies, and training machine learning algorithms introduces the phenomenon of underspecification. These observations imply that every task is associated with a space of solutions. To date, the structure of this space is not understood, limiting the approach of comparing RNNs with neural data. Here, we characterize the space of solutions associated with various tasks. We first study a simple two-neuron network on a task that leads to multiple solutions. We trace the nature of the final solution back to the network's initial connectivity and identify discrete dynamical regimes that underlie this diversity. We then examine three neuroscience-inspired tasks: Delayed discrimination, Interval discrimination, and Time reproduction. For each task, we find a rich set of solutions. One layer of variability can be found directly in the neural activity of the networks. An additional layer is uncovered by testing the trained networks' ability to extrapolate, as a perturbation to a system often reveals hidden structure. Furthermore, we relate extrapolation patterns to specific dynamical objects and effective algorithms found by the networks. We introduce a tool to derive the reduced dynamics of networks by generating a compact directed graph describing the essence of the dynamics with regards to behavioral inputs and outputs. Using this representation, we can partition the solutions to each task into a handful of types and show that neural features can partially predict them. Taken together, our results shed light on the concept of the space of solutions and its uses both in Machine learning and in Neuroscience.

## 1 Introduction

Modern machine learning operates in an over-parameterized regime, implying that many different parameter-sets can achieve low error on a given training set (1). This observation implies that for every task, there exists a space of solutions that can implement it. What are the properties of such a solution space? Are networks able to learn solutions that capture the intended underlying

35th Conference on Neural Information Processing Systems (NeurIPS 2021).

phenomena or do they reach artificial shortcuts that do not generalize well? What biases networks to prefer one solution over the other? These questions remain largely unanswered. A parallel phenomenon occurs in Neuroscience. When animals are instructed to perform a task in a controlled environment, they exhibit both neural and behavioral variability, which stem from different task-strategies (2; 3; 4; 5; 6; 7; 8; 9; 10; 11). In Computational Neuroscience trained Recurrent Neural Networks (RNNs) are used as a tool to explain functions and mechanisms that are observed in brain dynamics (12). In fact, various recent studies have matched the activity of trained RNNs to that of experimental recording (13; 14; 15; 16; 17; 18; 19). In light of the variability that undoubtedly exists on both sides of the comparison, these results seem puzzling. In this work, we present multiple tasks for which trained RNNs produce a rich space of qualitatively different solutions. We argue that to properly use artificial networks, and RNNs in particular, as models of neural circuits that support a given task, it is necessary to chart the space of solutions that arises from training.

Here, we first apply this approach to a simple two-neuron network and demonstrate how distinct solutions arise. We then study three tasks inspired by the neuroscience literature: interval reproduction (20), delayed discrimination (21), and interval discrimination (22). We show that different networks with identical hyperparameters find qualitatively different solutions. We find one layer of variability within the neural activity in response to stimuli from the training set. Since by design the output of all networks is identical during training, this layer is akin to multiple realizability (23). Next, we expose an additional layer of variability when we challenge the networks with inputs that are outside the distribution of the training set and systematically characterize the responses. Furthermore, we manage to show that the diversity revealed with these challenging inputs corresponds to qualitatively different computations performed by the network. To chart the space of solutions, we introduce a tool that reduces the dynamics of a network into a graph that captures the essence of the computation performed. Applying it to all networks partitions the space into a handful of possible reduced dynamics. Additionally, these classes can be partially predicted using experimentally accessible neural activity obtained only in response to trained stimuli.

## 2 Related work

Recent work, (24) studied the effects of modeling choices over the dynamics of the trained solutions, in a few canonical tasks. They trained thousands of RNNs, while systematically controlling the hyperparameters, and analyzed the geometrical and topological aspects of each solution. They concluded that while the geometry of different solutions can vary significantly across different architectures, the underlying computation and dynamical objects are widely universal. Our results are superficially opposite to theirs. This is probably due both to the choice of tasks and to challenging networks with unexpected inputs.

The authors of (25) show that deep networks trained on vision tasks develop different internal representations, especially in the higher layers. While reaching similar conclusions on the need to use populations of networks for neuroscience comparisons, the analysis of feedforward networks naturally focuses on representations rather than on dynamics. While writing this manuscript, we were made aware of recent work by (26) that takes a very similar approach to ours. Apart from the different tasks, architectures and learning rules studied there, our analysis of the two-neuron network also provides a tangible example of discrete basins of attraction in solution space. An underspecified two-dimensional epidemiological model was studied by (1). The variability in solutions there, however, was continuous. Specifically, all solutions were connected on a single manifold and the dynamics of the system did not undergo bifurcations along this manifold.

Underspecification was also highlighted in neuroscience models that are not based on artificial neural networks (27) and is observed more generally in complex systems (28).

## 3 What constitutes a solution?

Before comparing how the different networks solve the various tasks, it is worth dwelling on the concept of a solution. This is not a trivial concept and can be related to fundamental philosophical questions. Aristotle suggested (29) the four causes of understanding an object: its material form, its formal description, its efficient cause (creation process), and its final cause (purpose). We can draw an analogy to our understanding of RNN solutions, and ask: What is their architecture (material),

what is their underlying algorithm or dynamics (formal), which optimization process led to their final state (efficient), and what task do they solve (final). Because our motivation stems from comparing networks to biological data, we take an operative approach that relies on measures that could in principle be obtained experimentally. We thus consider two of these pillars: either the neural activity of the network while performing the task (formal description), or its predictions for unexpected stimuli (purpose).

## 4 Space of solutions in a 2D RNN task

We first study the space of solutions in an extremely simple scenario that nevertheless shows qualitatively different solutions. We consider a 2-neuron continuous-time network (Figure 1A), in which the state of the network $\boldsymbol{x} \in \mathbb{R}^2$ evolves according to:

$$\dot{\boldsymbol{x}} = -\boldsymbol{x} + W\phi(\boldsymbol{x}), \; \boldsymbol{x}(0) = \begin{bmatrix} 1 \\ 0 \end{bmatrix} \tag{1}$$

For the task, we require that $\boldsymbol{x}(T) = \begin{bmatrix} 0 \\ 1 \end{bmatrix}$ (Figure 1B). We choose $\phi := \tanh$ and $T = 10$. Accordingly we define a loss:

$$\mathcal{L} = \frac{1}{2} \left\| \boldsymbol{x}(T) - \begin{bmatrix} 0 \\ 1 \end{bmatrix} \right\|^2. \tag{2}$$

The system has four parameters, given by the elements of $W \in \mathbb{R}^{2 \times 2}$. We train 10000 randomly initialized networks to minimize $\mathcal{L}$ and discover solutions with qualitatively different dynamics. We select three representative examples. (Figure 1 D,E,F). The first example implements a stable fixed point near the target, the second implements a limit cycle near the target, and the third exhibits transient amplification passing through the target before gradually decaying to zero.

To visualize the space of solutions, we consider a 2D plane in the 4D parameter space containing the three aforementioned solutions. We find that this plane explains 94 percent of the variance of all solutions (compared to 95 percent by the first two principal components), and is thus a representative description of solution space. The shading in Figure 1C shows the loss along this plane, indicating that some solutions lie along a single continuous manifold, while others inhabit discrete islands. Linearizing the dynamics at the origin allows us to obtain a bifurcation diagram on this plane (Figure 1C, red and black lines), showing the existence of a bifurcation along the continuous manifold. In the simpler case of a linear network, we can also show how the discrete and continuous solution sets arise (see supplementary section 2.1)

We thus conclude that the network converges to qualitatively different solutions. These can be attributed to discrete basins of attraction in parameter space and dynamical bifurcations of the system occurring within and across these basins.

## 5 Neuroscience Tasks

Guided by the results from the two-neuron network, we examine three more complex, neuroscience-inspired, tasks Figure 2. A priori, it is not clear whether such tasks will exhibit more or less variable solutions. On the one hand, a complex task might lend itself to multiple algorithmic solutions. On the other hand, a complex task represents more constraints on the network and hence might lead to convergence to a unique solution. More details can be found in the supplementary material subsection Section 2.1.2

In the **Delayed discrimination** task (21), two pulses of varying amplitudes ($f_1, f_2 \in [2, 10]$) are presented, separated by a varying delay ($t_d \in [5, 24]$) Figure 2A. The lower panel shows the correct output in response to various stimulus combinations, which is independent of $t_d$ and partitions the $(f_1, f_2)$ plane.

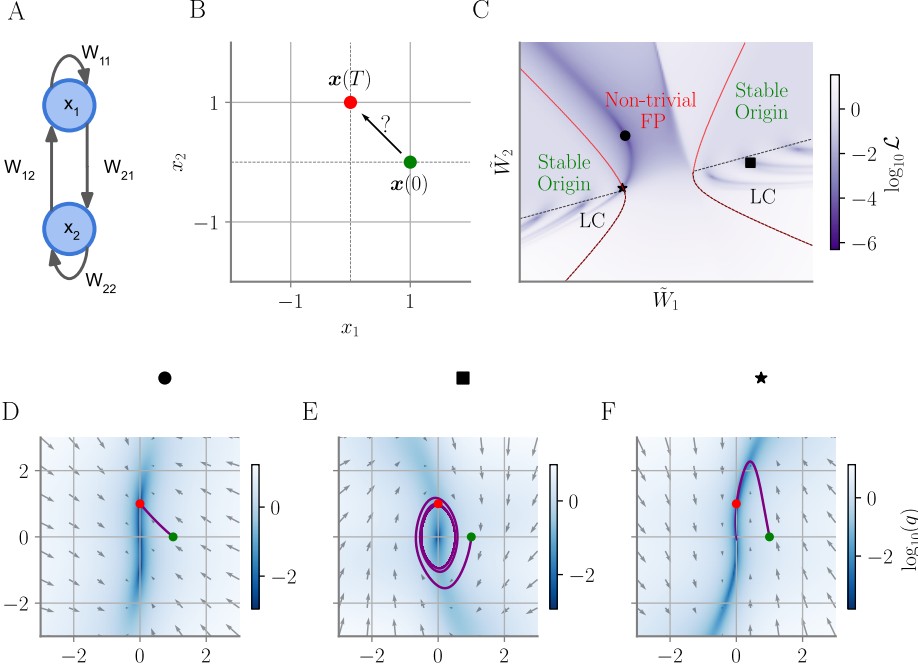

Figure 1: Two-neuron network. **A** Network architecture, with the four trainable parameters. **B** The task defined in phase space. The initial state is fixed (green). The task is for the network to be at a final state (red) at time $T = 10$. **C** A 2-D slice of the 4-D parameter space containing three selected solutions (black markers). Heatmap denotes task loss evaluated along the slice. Overlaid is a bifurcation diagram along the slice; lines indicate dynamical bifurcations and text indicates regions with a stable origin, non-trivial fixed point attractors and limit cycles (LC). **D, E, F** Phase portraits of the three selected solutions marked in **C**, trajectories taken by each of the networks during the task ($t < T$, thick purple) and subsequently ($t > T$, thin purple). Heatmaps denote speed of dynamics $q := \frac{1}{2} \|\dot{\boldsymbol{x}}\|^2 (30)$

.

In the **Interval discrimination** task (22), two pulses of equal amplitude are presented at times $t_1$ and $t_1 + t_2$, where $t_1, t_2 \in [10, 30]$ and $t_1 \neq t_2$. The network should produce an output pulse whos sign indicates whether $t_2 > t_1$ or not Figure 2B. The lower panel once more shows the desired output.

Finally, in the **Interval reproduction** task (20) the network receives two input pulses – Ready and Set – separated by $t_{in}$ time steps. The task is to generate an output pulse $t_{out} = t_{in}$ time steps after the Set pulse. The training intervals were drawn from a uniform distribution $t_{in} \in [20, 50]$. The lower panel shows the desired output, where having only one parameter defining the trials ($t_{in}$) allows us to display the entire trial output on each line. Trials aligned to the *Ready* pulse (Figure 2B, red). This results in the *Set* pulse (yellow) forming a line with slope 1.0 and the *Go* pulse (green) a line with slope 0.5. Figure 2C shows the output of a trained network matching this pattern.

For each task we trained 400 Vanilla networks, with $N = \{20, 30, 40, 50\}$ hidden units. See the Supplementary material subsectionSection 2.1.3 for more details regarding the training process, as well as results from GRU and LSTM networks trained on the interval reproduction task.

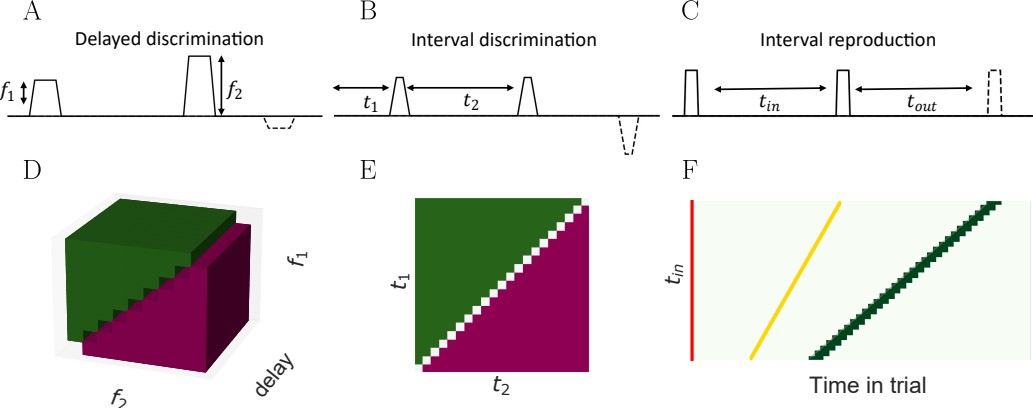

Figure 2: The three tasks. **A** Delayed discrimination task. Top: The network is presented with two pulses with amplitudes $f_1$ and $f_2$ respectively, separated by a variable delay, and should respond with a pulse with an amplitude equal to $\text{sign}(f_1 - f_2)$. Bottom: The desired output for all training data, for each delay, $f_1$, $f_2$ the color of the 3D matrix is the expected response. **B** Interval discrimination task. Top: The network is presented with two pulses with a unit amplitude, that arrive at times $t_1$ and $t_2 - t_1$. In response, it should produce a pulse with amplitude that is equal to $\text{sign}(t_2 - t_1)$. Bottom: Desired output for all $t_1$ and $t_2$ combinations. **C** Interval reproduction (Ready-Set-Go) task. Top: The network is presented with two pulses (Ready and Set, provided through separate input channels) separated by $t_{in}$ time-steps. The desired output is a Go pulse, delayed by $t_{out} = t_{in}$ steps from the Set pulse. Bottom: Desired output for all trials, with Ready, Set and Go depicted in red, yellow and green respectively..

## 5.1 Multiple neural dynamics solve the same task

We demonstrate the diversity of neural dynamics in different solutions by showing a couple of examples from each task. The left side of Figure 3AB shows the activity of two networks solving the interval reproduction task projected onto the first principal components. It is possible to infer two distinct algorithms for solving the timing problem from these plots. Because the task requires counting time in two distinct epochs – from Ready to Set and from Set to Go – we focus on these to explain the algorithms. The network in panel B uses the same phase space trajectory for both epochs, effectively counting from the maximal delay ($t_{in} = 50$) downwards upon the Ready pulse, and moving to an earlier point upon the Set pulse. This is in contrast to the network in panel A, in which the Set pulse leads to a different area in phase space. Furthermore, the almost circular trajectory of the Ready-Set epoch hints at the oscillatory behavior that will be discussed below.

The delayed discrimination task also admits multiple solutions. In this case, we consider activity during the delay period. Figure 3CD show several delay trajectories, corresponding to different $f_1$ values. The network in panel D converges to approximate fixed points for each such value, while the network in panel C converges to limit cycles. Note that the network was trained with variable delays (31), and hence the second stimulus arrives at random phases of these cycles. Nevertheless, both networks perform well on the training set.

Interval discrimination contains two timing-epochs – from the onset of the trial to the first pulse, and between the two pulses. The network in panel E uses the same trajectory in phase space for both epochs, similar to panel A of the interval reproduction task. The second example exhibits both oscillatory behavior and distinct trajectories for the two phases.

Taken together, these examples show that networks trained with identical hyperparameters, and reaching similar performance nevertheless develop qualitatively different neural dynamics to solve the same task.

## 5.2 Extrapolation

On the behavioral side, we challenge networks with stimuli that are outside of the trained distribution. Since each of our tasks is parametric, testing the networks on extrapolation while increasing each parameter is a natural choice. The middle column of Figure 3 shows the results of this challenge for the various networks, which often shows traces of the neural diversity described above.

The almost circular Ready-Set neural trajectory of panel A apparently results in a limit cycle, as revealed by the extrapolation plot. Similarly, the fact that the Ready-Set trajectory of panel B leads directly to the Go pulse is reflected in the vertical Go line that precedes the Set pulse in the extrapolation plot. Other features, such as the fact that a Set pulse delivered *after* the Go pulse results in a second output, are only revealed through extrapolation and cannot be deduced from the PCA of neural activity during the training set.

In the delayed discrimination task, the oscillations shown in panel C were not reflected in the output on the training region. Extrapolating to larger amplitudes and delays, however, reveals output oscillations as depicted on the top face of the extrapolation cube.

In the interval discrimination task (panels E,F), the relation between neural trajectories and extrapolation behavior is less clear. But in this case, as in all others, the striking differences between different solutions are manifested both in the training neural activity and in the response to behavioral challenges.

## 5.3 Reduced Dynamics

Part of the motivation for looking into extrapolation patterns is uncovering the algorithm that networks use to solve the task. Within the framework of neural dynamics (18; 12) this corresponds to mapping the dynamical objects used by the network and their relation to behavioral inputs and outputs. Previous works focused on fixed point topology to achieve this goal (30; 24; 14). The tasks considered in the present work are mostly dependent on transient dynamics and hence require a different approach. To this end, we developed a tool to compress the state space while preserving the essential dynamics and the behavioral information. Briefly, the method generates trajectories by combining long stretches of autonomous dynamics with behavioral inputs. The trajectories are then merged and compressed into a graph. Each node represents areas in phase space that have a given output and share the same input-dependent past and future. The edges represent autonomous dynamics (black) or the various inputs (colors). The full details of the algorithm are in the supplementary material subsection 2.2. The left column of Figure 3 shows such reduced dynamics for all the example networks. Examining these graphs can reveal the dynamics driving extrapolation patterns.

The network of panel A has a limit cycle, which is depicted by the red node that has a black self-loop. Furthermore, the graph shows that after the Go pulse, the network can still respond to an additional Set pulse. The fact that the network of panel B utilizes the same trajectory for both epochs is reflected in the yellow self-loop in the corresponding graph. The graph also shows that a Go pulse can occur before the Set pulse.

The delayed discrimination task has more input types and therefore was less compressed by the algorithm. As a result, the graphs can show both the logic of comparison during the training set and the extrapolation patterns with the limit cycles. A Similar scenario holds for the interval discrimination task.

## 5.4 A space of solutions

The examples described above suggest that similar to the two-neuron case, trained networks converge into distinct solutions which can be characterized both by neural signatures and by behavioral ones. In the case of the two-neuron network, the parameter space is only four-dimensional, and most of the solutions are in a two-dimensional subspace. Hence, we could span most of it and understand the solution types via the linearized dynamics around the origin. In the more complex tasks, the reduced dynamics graphs serve as a method to characterize the space of solutions. We computed the reduced dynamics for all 400 networks of each task and found that they only contain a few different graphs. Figure 4E shows the number of networks of each type for the interval reproduction task. The four major graphs and their corresponding representative extrapolation plots are shown in panels A-D.

## 5.5 Inferring reduced dynamics from neural activity

The reduced dynamics described above are obtained by simulating the networks for very long times, to obtain their asymptotic behavior. In a neuroscience setting, such information is not readily available. Is it possible to infer this asymptotic behavior from neural activity during the training set? To answer this question, we extracted a set of features, as detailed in the supplementary material. The features mostly measure the relationship between neural activity in the different epochs of the task. Other features measure geometrical properties of a single epoch, as in the curvature of the Ready-Set trajectory, which was inspired by networks like Figure 3A. While we tried to include features that are natural descriptors of the neural activity, there is some arbitrariness in our choice. We chose to err on the side of including more features, and later rely on cross-validation to avoid overfitting in our predictions (32).

For each task, we trained and evaluated a classifier to predict the reduced dynamics of the pool of networks from the neural features. Figure 4F shows the resulting confusion matrix for the interval reproduction task (other tasks are in the supplementary material Figures 8 and 9). Cohen's kappa (33) was used to summarize the prediction quality, showing an above-chance performance.

## 5.6 Different architecture and tasks

The diversity of solutions reported here arise from identical hyperparameters and random initialization. To probe the biases introduced by changing hyperparameters, we tested the effect of varying network architecture for the interval reproduction task. The supplementary material shows results from training this task using LSTM and GRU networks. We find that some solution classes are shared between architectures, and some only appear in certain architectures (Figure 5A). Furthermore, these choices bias the solutions, as reflected in the different histograms. For instance, the example of Figure 5B did not occur in any of the 400 Vanilla networks, but was the most common type in GRU networks. Despite these biases, there are solutions that appear under all architectures, as exemplified in Figure 5C,D,E.

The chosen tasks so far emphasized variability. We also examined a context-dependent integration task, which was previously shown to exhibit a universal solution (24). The supplementary material shows an analysis of the response of networks to various behavioral challenges. We find that there is considerable variability in the response of networks to challenges, but of a quantitative rather than a qualitative nature. Furthermore, we also quantified various aspects of neural variability in these networks. We did not find significant correlations between the various behavioral and neural measures, suggesting that there are many axes of individuality for these networks.

## 6 Discussion

In recent years, trained RNNs were successfully used to model biological circuits. Specifically, these networks converged to solutions that were similar to those of their biological counterparts. This observation is puzzling for any complex system and particularly regarding the brain, a complex biological system in which variability is the rule rather than the exception. Inspired by this puzzle, we study the space of solutions of RNNs for both a simple two-neuron network and three neuroscience-inspired tasks. Through an interplay between observation, analysis, and perturbation in the form of extrapolation, we discovered for each task qualitatively different solutions. By analyzing the neural activity of these networks, we observed that the variety of behavioral phenotypes originate from only a handful of distinct dynamical mechanisms. To affirm these observations and describe the essence of each solution, we introduced a tool that summarizes the network computation in a reduced-dynamics graph that contains the essence of the dynamics as they relate to behavioral inputs and outputs. By extracting neural features from the training activity, we showed that the solution type can be partially predicted from experimentally accessible measurements.

We show using a 2D continuous-time RNN that qualitatively different dynamical topologies can arise in the context of a single task and that learning can find different solutions depending on initialization. In a related study, the authors analyze a low-D continuous-time GRU and find a large diversity of dynamical topologies (34). They find that the 2D GRU finds topologically different solutions to variants of a working memory task depending on the input statistics. Such a modification to the task, although subtle, modifies the loss landscape and learning dynamics. In our work we emphasize

that such topological diversity of solutions can arise for the same task, that is to say within the same loss landscape. Both our work and theirs identify the implications of topological diversity for generalization.

Asking a seemingly trivial question - "what is a solution?", highlights a complex and vital topic. Even though artificial and biological networks are being compared regularly, and there is a common intuitive understanding of what properties are relevant for this comparison, the precise definitions of "solution" or "mechanism" are rarely discussed (23). Is dynamical similarity within the training regime sufficient for asserting two solutions are alike? What is the proper way to test subjects' performance on a learned task? Can we answer these questions while considering the specifics of each task? Under what conditions can we safely assume that there exists a universal solution? We believe that every study that models neural circuits should be explicit about such meta-concerns. In this work, we charted solutions with an operational approach in mind and considered solutions as different as long as they qualitatively differ either in their neural activity within the training set or in their behavioral prediction on extrapolation trials. The reduced dynamics tool is our attempt to describe and differentiate solutions according to the criteria that we consider important.

Our work has several limitations that should be noted. We focused on the variability arising with identical hyperparameters and therefore did not systematically explore the effect of changing them. Specifically, how network size, learning algorithm, and choice of architecture bias the solutions remains to be explored.

The space of solutions for a given task is, above all else, a property of the task itself. We showed examples of three tasks that have substantial variability, and of one task (context dependent integration) that has much less variability. Yet, we do not know why some tasks admit multiple solutions and others do not. We conjecture that the tasks presented in (24) require an informative output at all points in time, whereas the tasks we considered require an informative output only once at the end of the trial.

The reduced-dynamics tool relies on a set of trajectories as a starting point. We opted to include regions in phase space that are reachable with task-consistent inputs (similar to (30)). While exploring the full phase space is not feasible, our choice may limit the description obtained by our tool. Finally, the space of solutions was described for networks that have already learned to perform a task. It remains to be seen whether it can be used as a map in which to understand how the process of learning itself takes place.

To conclude, we found that RNNs can produce a diverse set of solutions to the same computational tasks. These solutions represent distinct algorithms and are supported by corresponding dynamical objects. The solutions are isolated in parameter space, causing the initial conditions to largely determine the outcome. Furthermore, we showed that experimentally accessible tools can be used to indirectly characterize the asymptotic properties of the solutions. We believe that exploring the space of solutions can advance neuroscience, machine learning, and their intersection – making more rigorous comparisons of models to data.

## Acknowledgments and Disclosure of Funding

This work was supported in part by the Israeli Science Foundation (grant number 346/16, OB) and by a Rappaport institute thematic grant (OB).

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

Figure 3: Diverse solutions for the same tasks. Two example networks are shown for each task (rows). Left column: two-dimensional PCA of the network activity. Middle column: Network output in response to extrapolation across task parameters. Right column: the reduced dynamics of the networks. **(A,B)** Solutions to the time production task. The PCA plots show the activity from the Ready pulse (red dot and black line), as well as from the Set pulse (yellow) up to the beginning of the output production, for three different task parameters (colorbar). Note that in A the Ready-Set epoch is separated from the Set-Go, whereas in B these epochs converge. **(C,D)** Solutions to the delayed discrimination task. The PCA plots show the activity between the two input pulses, for various $f_1$ amplitudes. The memory of the different amplitudes is kept either via limit cycles (C) or slow points (D). **(E,F)** Solutions to the interval discrimination task. The PCA plots show activity from trial onset (black dot and line), as well as from the first pulse (yellow) until the maximal delay, for three different $t_1$ values (colorbar). Note the convergence of the two epochs to the same trajectory in E, similar to B.

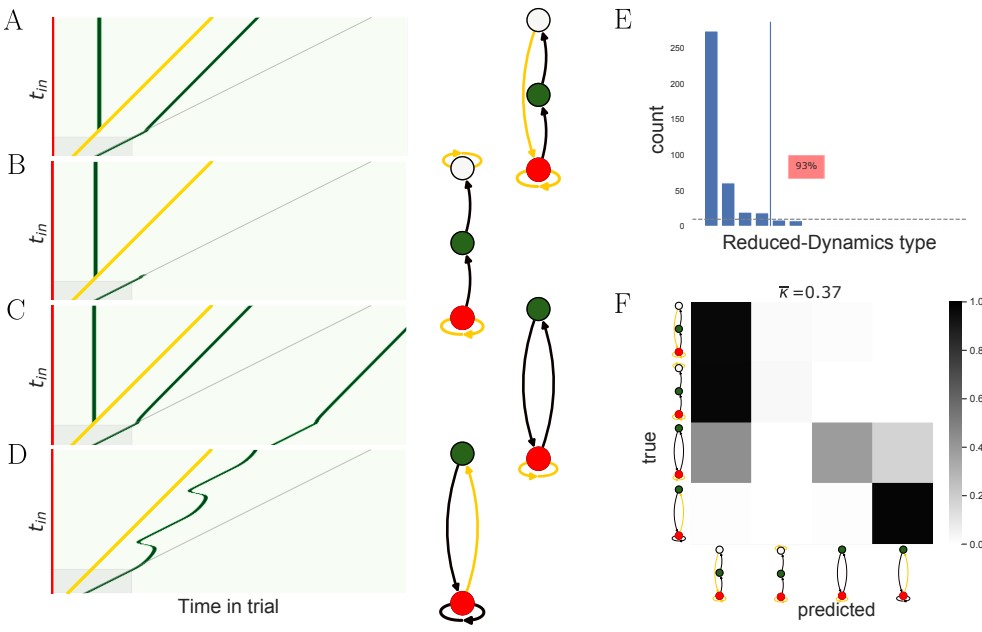

Figure 4: The space of solutions for the interval reproduction task. **A-D** Representative extrapolation plots (left) and reduced dynamics graphs (right) for the four most common solution types. **E** Distribution of solution types for the 400 networks trained. The four solutions shown account for 93% of the networks. *F* Neural features obtained during the training set can partially predict the solution type that includes extrapolation dynamics. The confusion matrix shows the result of this prediction. Note that the first two solution types are mixed by this prediction, but their dynamics during the training intervals is similar and they only differ in the dynamics after the Go pulse.

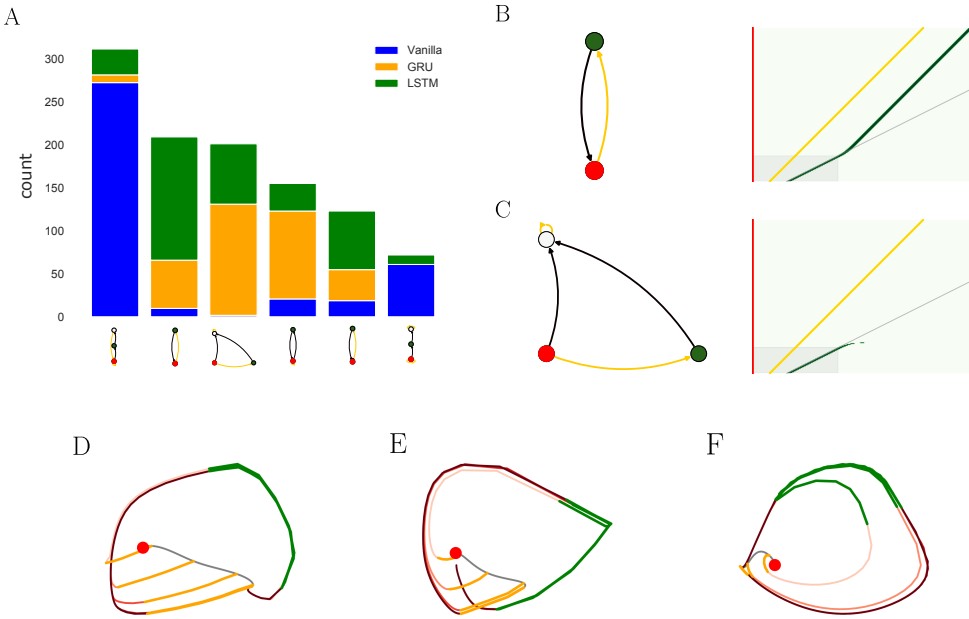

Figure 5: Architecture biases, but does not determine the solution. **A** A histogram of the six most common reduced dynamics across all three architectures for the time-reproduction task, shown by stacking the architecture-specific histograms on top of one another. **B, C** The reduced dynamics (left) and the extrapolation patterns (right) of the second and third most common solutions across all architectures, but rarely occur within the Vanilla networks. **D, E, F** The 2D PCA of the dynamics of three networks from all three architectures (Vanilla, GRU, LSTM), that reach the solution shown in panel **B**.

