# CHARTING AND NAVIGATING THE SPACE OF SOLUTIONS FOR RECURRENT NEURAL NETWORKS SUPPLEMENTARY MATERIAL

**Elia Turner**
Department of Mathematics
Technion, Israel Institute of Technology
eliaturner@campus.technion.ac.il

**Kabir Dabholkar**
Department of Mathematics
Technion, Israel Institute of Technology
kabir@campus.technion.ac.il

**Omri Barak**
Rappaport Faculty of Medicine and Network Biology Research Laboratory
Technion, Israel Institute of Technology
omri.barak@gmail.com

A python code for the figures and results is partially available at `https://github.com/eliaturner/space-of-solutions-RNN/`.

## 1 Two-neuron RNN

### 1.1 Discrete and continuous solution manifolds in the linear two-neuron RNN

In this section, we analyze a simplified linear discrete-time version. This allows an analytical solution of the weights required to perform the task and demonstrates the nature of the space of solutions. The equations are now

$$\boldsymbol{x}_{t+1} = W\boldsymbol{x}_t \tag{1}$$

$$\boldsymbol{x}_0 = \begin{bmatrix} 1 \\ 0 \end{bmatrix} \tag{2}$$

where $W \in \mathbb{R}^{2\times 2}$.

The task requires that $x_n = \begin{bmatrix} 0 \\ 1 \end{bmatrix}$. This can be solved either analytically (below) or numerically by minimizing the following loss by running gradient descent on $W$ from an initial $W_0$:

$$\mathcal{L} = \left\| x_n - \begin{bmatrix} 0 \\ 1 \end{bmatrix} \right\|^2 \tag{3}$$

We observe that different random initializations converge to different points in the space of solutions. In this document, we try to describe and visualize this space.

35th Conference on Neural Information Processing Systems (NeurIPS 2021).

### 1.1.1 The space of solutions

We need solutions to the following equations:

$$\begin{bmatrix} 0 \\ 1 \end{bmatrix} = A \begin{bmatrix} 1 \\ 0 \end{bmatrix} \tag{4}$$

$$A = W^n \tag{5}$$

This breaks down the problem into a feedforward problem of solving (4) for $A$ and a recurrent problem of solving (4) and (5) for W.

The space of solutions to the feedforward problem is simple:

$$\mathcal{A} := \{A \in \mathbb{R}^{2 \times 2} \text{ such that } A_{11} = 0 \text{ and } A_{21} = 1\}. \tag{6}$$

To solve the recurrent problem, we need to additionally find the $n^{\text{th}}$ real roots of $A \in \mathcal{A}$ if they exist. For simplicity, we focus on a task of only two time steps $n = 2$.

### 1.1.2 Solutions of a two time step task

We begin by considering only nonsingular $A$ ($A_{12} \neq 0$ in this case). The existence and number of real $W$ (branches), such that $A = W^2$ is given by spectrum of $A$. If $A$ has any real negative eigenvalues, then there are no real solutions. If $A$ has no real negative eigenvalues then $A$ has $2^{r+c}$ square roots where $r$ is number of real eigenvalues and $c$ is the number of complex eigenpairs ((**?** ) Theorem 7).

First we study the case of $A$ with complex eigenvalues $\theta \pm i\mu$ with $\mu \neq 0$.

$$\mathcal{A}_{\mathbb{C}} = \{A \in \mathcal{A} \text{ such that } \operatorname{Tr}(A)^2 - 4\det(A) < 0\} \tag{7}$$

$$= \{A \in \mathcal{A} \text{ and } A_{22}^2 + 4A_{12} < 0\} \tag{8}$$

$A \in \mathcal{A}_{\mathbb{C}}$ has $2^1$ real roots given by $W = \pm(\alpha\mathbf{1} + \frac{1}{2\alpha}(A - \theta\mathbf{1}))$ where $(\alpha + i\beta)^2 = \theta + i\mu$. (Eq 4.6 in (**?** ))

Next we identify the the $A$s with eigendecomposition $A = V \begin{bmatrix} \theta_1 & 0 \\ 0 & \theta_2 \end{bmatrix} V^{-1}$ where the $\theta_i$s are real and positive.

$$\mathcal{A}_{\mathbb{R},>0} = \{A \in \mathcal{A} \text{ such that } \operatorname{Tr}(A)^2 - 4\det(A) > 0 \text{ and } \operatorname{Tr}(A) > \sqrt{\operatorname{Tr}(A)^2 - 4\det(A)}\} \tag{9}$$

$$= \{A \in \mathcal{A} \text{ and } A_{22}^2 + 4A_{12} > 0 \text{ and } A_{22} > 0 \text{ and } A_{12} < 0\} \tag{10}$$

There are $2^2$ real roots of $A \in \mathcal{A}_{\mathbb{R},>0}$ given by $W = V \begin{bmatrix} \pm\sqrt{\theta_1} & 0 \\ 0 & \pm\sqrt{\theta_2} \end{bmatrix} V^{-1}$ taking each of the four combinations of the $\pm$s.

We numerically verify these statements by running gradient descent from random initialization $W_0^{(i,j)} \sim U(-3, 3)$. Figure 1 shows these solutions on the non-zero elements of the $A$ matrix. The left panel shows the continuous nature of the feed-forward problem. In contrast, the right panel shows a highly non-uniform distribution, matching the regions predicted from the analysis above.

We see that while the feedforward problem has a plane of degeneracy, the recurrent problem has two modes of degeneracy: firstly solutions fall somewhere in a subset of the same feedforward plane of degeneracy and secondly, at each point on the plane the solution can fall on one of several branches. While the first mode is continuous in nature, the second mode is disjoint.

### 1.1.3 Visualizing the Recurrent solutions

To visualize the branching of solutions in the recurrent problem we can plot them in 3D (Figure 2), where the added axis of $W_{11}$ allows us to see the multiple branches arising from square roots of $A$.

### 1.1.4 Dynamics in different regions

The above analysis indicated that the solution space can be divided into different regions. But are solutions from these different regions also characterized by different dynamics? Figure 3 shows this is indeed the case, by sampling solutions from the two regions and plotting their trajectories.

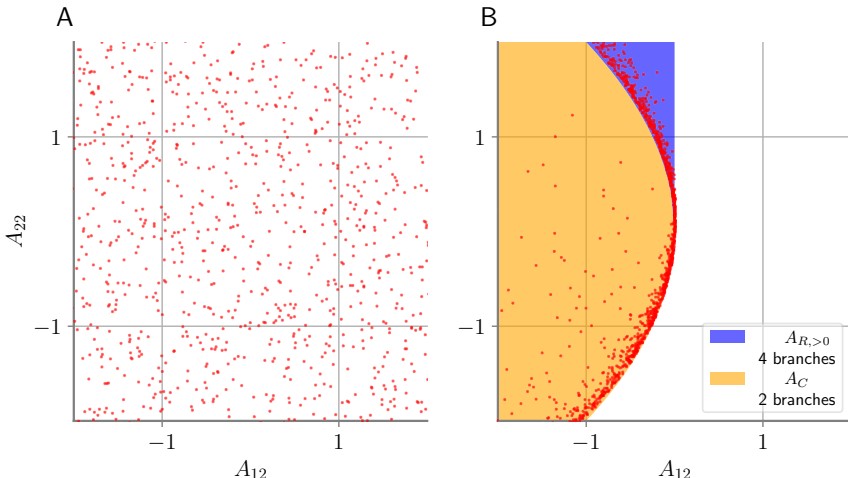

Figure 1: Solutions of gradient descent from random initialization to the feedforward problem (A) and the recurrent problem (B) in the space of $A$ matrices. We only plot the two unconstrained elements $A_{12}$ and $A_{22}$.

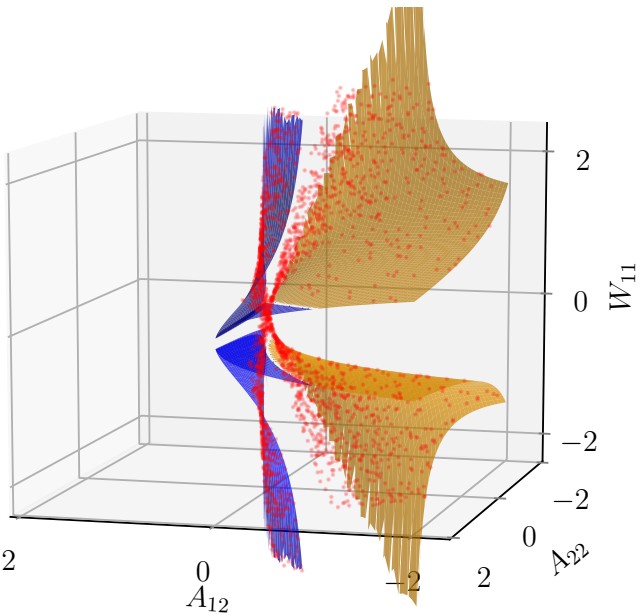

Figure 2: Space of solutions to the recurrent (n=2) problem in the space of the two unconstrained elements of $A$: $A_{12}$ and $A_{22}$ and one element of $W$: $W_{11}$. The surfaces are analytical solutions and dots are solutions obtained numerically by gradient descent from random initial conditions.

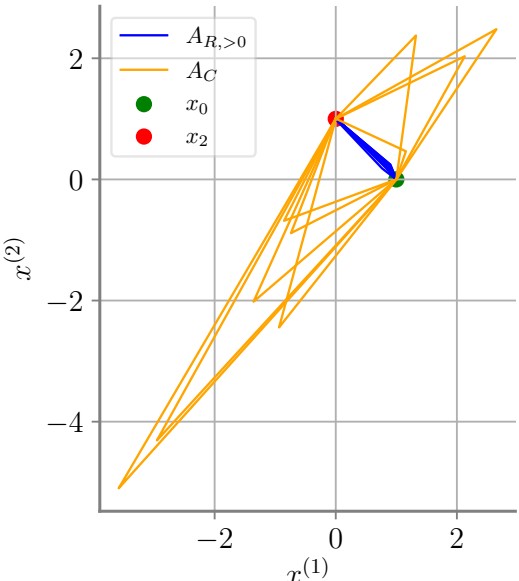

Figure 3: Trajectories of recurrent solutions (n=2 time steps) from the two classes $\mathcal{A}_{\mathbb{C}}$ and $\mathcal{A}_{\mathbb{R},>0}$.

## 1.2 Nonlinear two-neuron RNN details and ReLU version

### 1.2.1 Training methods

The four parameters, $W \in \mathbb{R}^{2\times 2}$, were iid sampled from a uniform distribution $\mathcal{U}(-1.5, 1.5)$. We implemented the continuous time dynamics of the RNN in *PyTorch* (1) using the package *torchdiffeq* (2; 3) enabling the calculation of gradients $\frac{\partial \mathcal{L}}{\partial W}$ using back-propagtion through time. We trained parameters using *Adam* (4).

### 1.2.2 Different nonlinearity

To examine the effect of training hyperparameters on the space of solutions, we used $\phi := \text{ReLU}$ instead of $\phi := \tanh$ that was used in the main text. We find that this choice indeed leads to different solution types. Specifically, ReLU RNNs did not converge to limit-cycle solutions. Some converged to non-zero fixed points, accompanied by a saddle point at the origin, as in the main text. In addition, two other solution types arose in this setting. A stable origin with large transient amplification, as in the yellow curve of Figure 4), and a diverging trajectory, shown by the dark curve in the same figure. The effect of the different non-linearity is also seen in the distribution of trace and determinants of the solutions (Figure 5), where limit cycles are absent for ReLU, and stable solutions (bottom-right quadrant) are present but uncommon for $\tanh$.

## 2 Neuroscience Tasks

### 2.1 Training process

### 2.1.1 Network architecture

We studied three different RNN architectures and their exact equations are all summarized below. The trained parameters are the weights $W$ and biases $b$. The function $\sigma(z) = (1 + exp(-z))^{-1}$ is the sigmoid function, $h_t \in \mathbb{R}^N$ and $u_t \in \{0, 1\}^2$ are the state and the input at time $t$.

**Vanilla (5)**

$$h_t = \tanh\left(W_{ih}u_t + b_{ih} + W_{hh}h_{t-1} + b_{hh}\right) \tag{11}$$

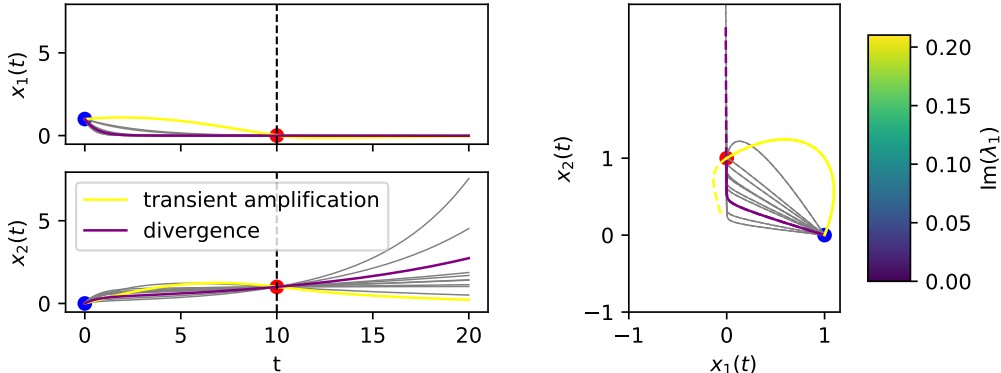

Figure 4: Trajectories of several 2D RNN solutions for $\phi := \text{ReLU}$.

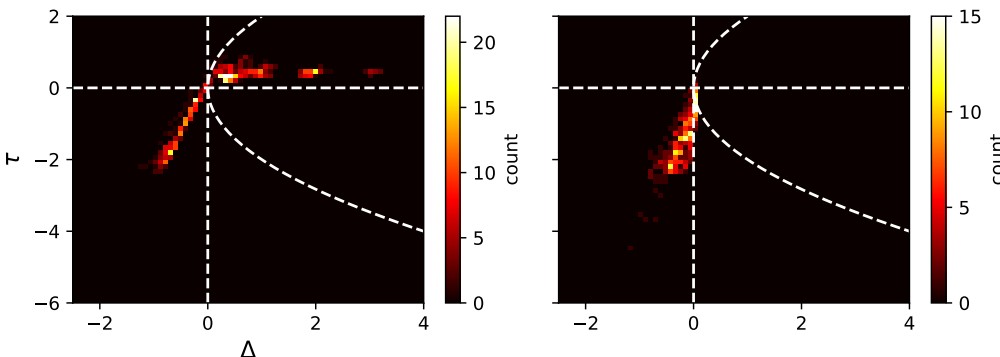

Figure 5: A 2D histogram of trace $\tau$ and determinant $\Delta$ of $W$ of the 2D RNN solutions for the task specified in the main text. $\phi := \tanh$ (left) and $\phi := \text{ReLU}$ (right).

**GRU (6)**

$$r_t = \sigma\left(W_{ir}u_t + b_{ir} + W_{hr}h_{t-1} + b_{hr}\right) \tag{12}$$
$$z_t = \sigma\left(W_{iz}u_t + b_{iz} + W_{hz}h_{t-1} + b_{hz}\right) \tag{13}$$
$$n_t = \tanh\left(W_{in}u_t + b_{in} + r_t * (W_{hn}h_{t-1} + b_{hn})\right) \tag{14}$$
$$h_t = (1 - z_t) * n_t + z_t * h_{t-1} \tag{15}$$

**LSTM (7)**

$$i_t = \sigma\left(W_{ii}u_t + b_{ii} + W_{hi}h_{t-1} + b_{hi}\right) \tag{16}$$
$$f_t = \sigma\left(W_{if}u_t + b_{if} + W_{hf}h_{t-1} + b_{hf}\right) \tag{17}$$
$$g_t = \tanh\left(W_{ig}u_t + b_{ig} + r_t * (W_{hg}h_{t-1} + b_{hg})\right) \tag{18}$$
$$o_t = \sigma\left(W_{io}u_t + b_{io} + W_{ho}h_{t-1} + b_{ho}\right) \tag{19}$$
$$c_t = f_t * c_{t-1} + i_t * g_t \tag{20}$$
$$h_t = o_t * \tanh(c_t) \tag{21}$$

The units had $N = 20, \ldots, 50$ hidden neurons and the output of the network at every time-step is an affine readout of the internal state. $h_0$ was always initialized to zero.

### 2.1.2 Task and trial structure

In all of the trials below, there are two input channels, one for each input pulse, and one output channel. Both inputs and the required output were binary sequences with ones during each pulse and

zero elsewhere. Each task is divided into seven epochs - before the first pulse, the first pulse, between pulses, the second pulse, before output pulse, output pulse, and after output pulse.

**Delayed Discrimination (8)**  The first pulse with amplitude $f_1 \in [2, 10]$ arrives after five steps. The second input pulse with amplitude $f_1 \in [2, 10]$,where $f_1 \neq f_2$ arrives after $5 + t_d$, where $t_d \in [0, 24]$. After 15 additional steps, the network is supposed to respond with a five-steps output pulse with amplitude sign$(f_2 - f_1)$.

**Interval Discrimination (9)**  The first pulse is given after $t_1$ steps, where $t_1 \in [10, 30]$. The second pulse is given $t_2$ steps after $t_1$. Both pulses last for two steps and have unit amplitude. After 15 additional steps, the network is supposed to respond with two-steps output pulse with amplitude sign$(t_1 - t_2)$.

**Interval Reproduction (10)**  The *Ready* pulse was given after $10 - 20$ steps. When working with intervals from the range $[t_{in}^{min}, t_{in}^{max}]$, the length of all trials was set to $2t_{in}^{max} + 100$. This allowed the network time to relax back to rest for at least 70 steps after emitting a *Go* pulse. All pulses were 5 steps long. For training $t_{in}^{min} = 20$ and $t_{in}^{max} = 50$

The training set always included 512 random trials so, on average, every interval was included more than 5 times.

### 2.1.3  Training protocol

All networks were trained using *Adam* (4) for 10000 epochs with a batch size of 64 and a decaying learning rate starting from $1e - 3$ up until $1e - 5$. Unless stated otherwise, the training set was comprised of 512 trials and their order was shuffled at the beginning of each epoch. We estimated the network's performance with mean squared error (MSE), and training was halted when the minimal threshold of $10^{-4}$ was achieved over the training set.

## 2.2  Reduced dynamics

As we discussed in the main text, defining what is a solution is not trivial. We follow the dynamical system approach (11; 12) and wish to obtain a compact description of the dynamics of the network and their relation to behavioral inputs and outputs. Previous work mostly focused on fixed points and their vicinity (13; 14). Because transient dynamics are at the heart of some of the tasks studied here, we opted for a different approach. To this end, we devised a tool that, given the network weights and task inputs, builds a directed graph that captures the essence of the calculation, which we call reduced dynamics. This process can be divided into two steps; representation of the dynamics as a directed graph, and pruning all irrelevant information from it. We will describe each of these steps next.

### 2.2.1  Dynamics to Directed Graph

Each set of dynamical trajectories and input-driven transitions between them can be interpreted as a directed graph, where each node holds a state and its corresponding readout as attributes, and each edge is weighted according to the input it represents. However, this representation is highly redundant because autonomous trajectories converge often to a lower-dimensional dynamical object. Hence, a more exact graph representation of the dynamics would recognize trajectories at the point of convergence as identical. Using this observation, we derived an iterative process to represent the dynamics of a network faithfully with a graph:

1. Create a graph from the autonomous trajectory of the network, from a chosen initial state.
2. Inject task inputs to states at the appropriate locations (see below) and obtain the immediate states afterward.
3. For each such state:
   (a) Run the dynamics from an initial state until a cycle is found or the neural speed is low.
   (b) Create a subgraph from that trajectory.
   (c) Connect the subgraph to the graph by the appropriate edge.
   (d) Try and merge the subgraph to other previously-existing branches.

There is a choice to make on how to collect candidate trajectories for this process. In principle, one could follow the autonomous dynamics, and from each state inject every possible input, and follow their outcome and so forth. Such a procedure can lead to an exponential increase in the number of trajectories and is thus not feasible. We chose to extend the autonomous parts of trajectories beyond those required by the task. The inputs, however, were only provided in a task-consistent manner. For example, in the interval reproduction task, we gave only one Ready pulse and one Set pulse for each *trial*. The product of this process is a directed graph that faithfully represents the dynamics and contains the least information possible.

**Branch-Merging**    The algorithm tries to match each new branch to the previously existing ones. If the branch contains a cycle, it would be compared to all existing cyclic components. Otherwise, it will be compared to non-cyclic components. In any case, for each such component, we estimate the convergence-score of each pair of states, according to the formula:

$$\text{score}(h_{t_1}^1, h_{t_2}^2) = \frac{2 \left\| h_{t_1}^1 - h_{t_2}^2 \right\|_2}{q(h_{t_1}^1) + q(h_{t_2}^2)}, \quad h_{t_1}^1, h_{t_2}^2 \in \mathbb{R}^n$$

where we quantified the neural velocity in phase space (13) as

$$q(h_t) = \| h_{t+1} - h_t \|_2, \quad t \in \mathbb{N} \tag{22}$$

Intuitively, the score measures the separation between trajectories using their velocity as the unit of measurement. A lower score indicates a better chance for convergence. By storing all such pairs in a matrix, we can look for sub-lower diagonals that contain values below a threshold as an appropriate candidate. In the end, the chosen branch will have the earliest connection point with the new branch. Importantly, the threshold can be tuned and used for different purposes - a lower threshold would generally lead to a higher resolution of the dynamics, and vice versa.

### 2.2.2   Full dynamics graph to a reduced dynamics graph

In this part, we remove parts from the graph iteratively until we reach an irreducible representation that contains the essence of the computation. To achieve this, we created a list of rules that operate on a local level - structures of up to four nodes. Each rule specifies a different condition under which a structure can be compressed, without causing information loss. The main rules are shown in Figure 6, and the exact implementation is deposited in the accompanying code. Stages of the process are shown for a specific network in Figure 7. Note that this is not a *lossless* compression; Some properties of the solution do not remain in the final form. For instance, the temporal distance between different states is eliminated. For the interval reproduction task, this does not allow to "see" the logic of the task from the graph, but it does allow to differentiate the major solution categories as described in the main text. The rules can be modified to include such information, and in general, the set of rules is flexible so that the user can define what type of information will remain in the final product.

### 2.3   Feature extraction

As described in the main text, we extracted various features from the neural activity from zero-input epochs within the training set. These were related to the major dynamical objects and task epochs. For each task, epochs corresponding to one- or two- dimensional manifolds were extracted as detailed below.

**Interval reproduction task**    We extracted the 1D trajectory between *Ready* and *Set* (until $t_{in}^{max}$) that is shared across trials, and a 2D manifold by extracting for each task parameter $t_{in}$ its following *Set-Go* trajectory and combining the results.

**Interval discrimination task**    We extracted the 1D trajectory during the first interval (until $t_1^{max}$) that is shared across trials, and a 2D manifold by extracting for each task parameter $t_1$ the activity during the interval that follows it and combining the results.

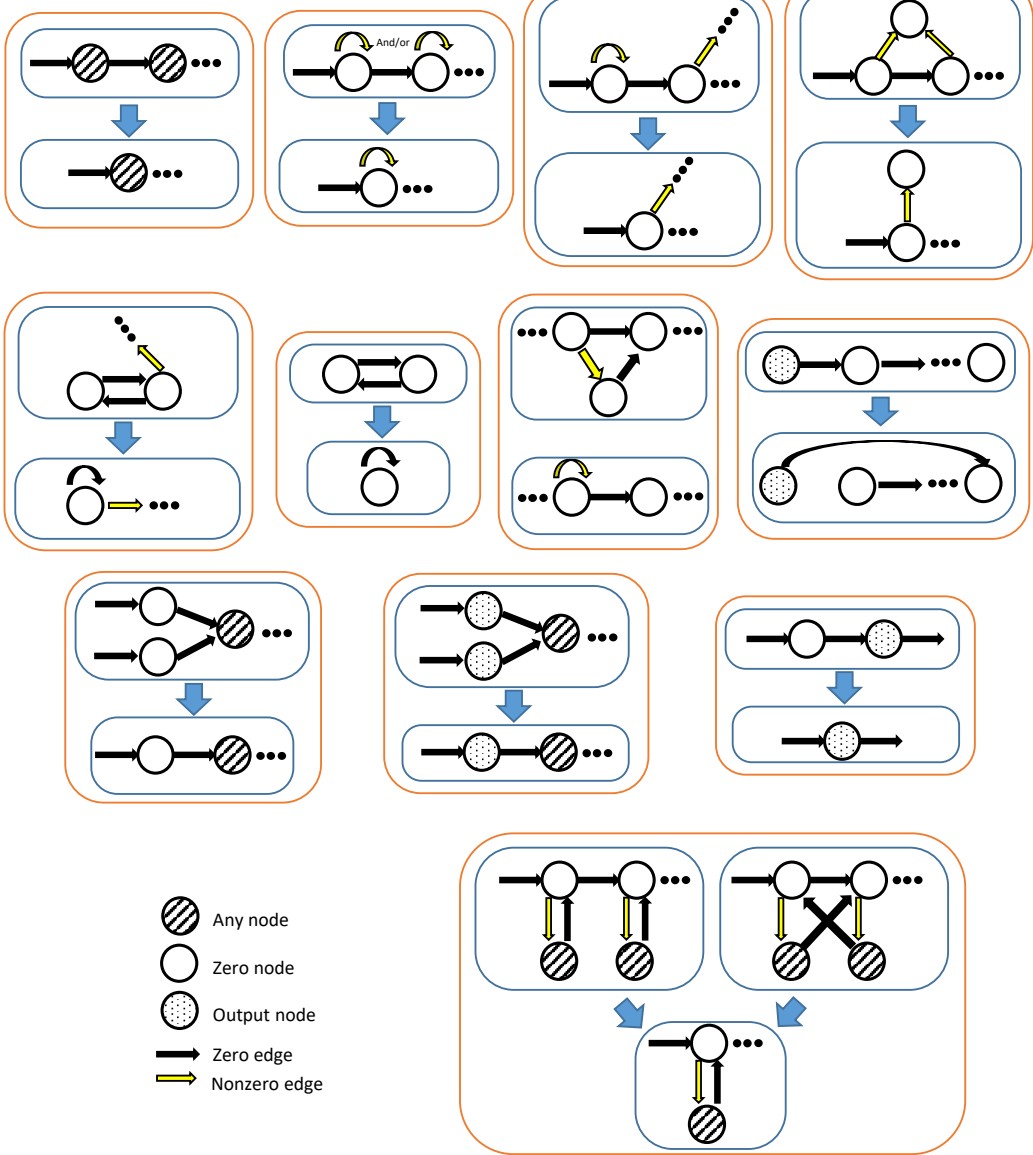

Figure 6: Reduced-dynamics rules

**Delayed discrimination** We extracted for each task parameter ($f_1$) the trajectory during the delay that follows it, and combined the results into a 2D manifold. The two trajectories between the second pulse and the output of the network corresponding to $(10, 2)$ and $(2, 10)$. In total one 2D object and two 1D objects. We extracted features from each 1D trajectory, from each 2D manifold, and similarity measures between pairs of 1D and 2D objects.

**1D features** The shape of a trajectory can indicate whether the network will eventually converge to a limit cycle. We thus considered the minimal and maximal curvature, the speed at its end, and the ratio between its initial and final speed. All these features were measured on a logarithmic scale.

**2D features** Because this is a 2D manifold (time by trials), we calculated the aspect ratio as follows. The nominator was the cumulative length of the trajectory corresponding to the initial states across all trials. The denominator was the length of the full trajectory of the longest trial. Similarly, we extracted the aspect ratio to the final states of the manifold. Later, we calculated how the length of

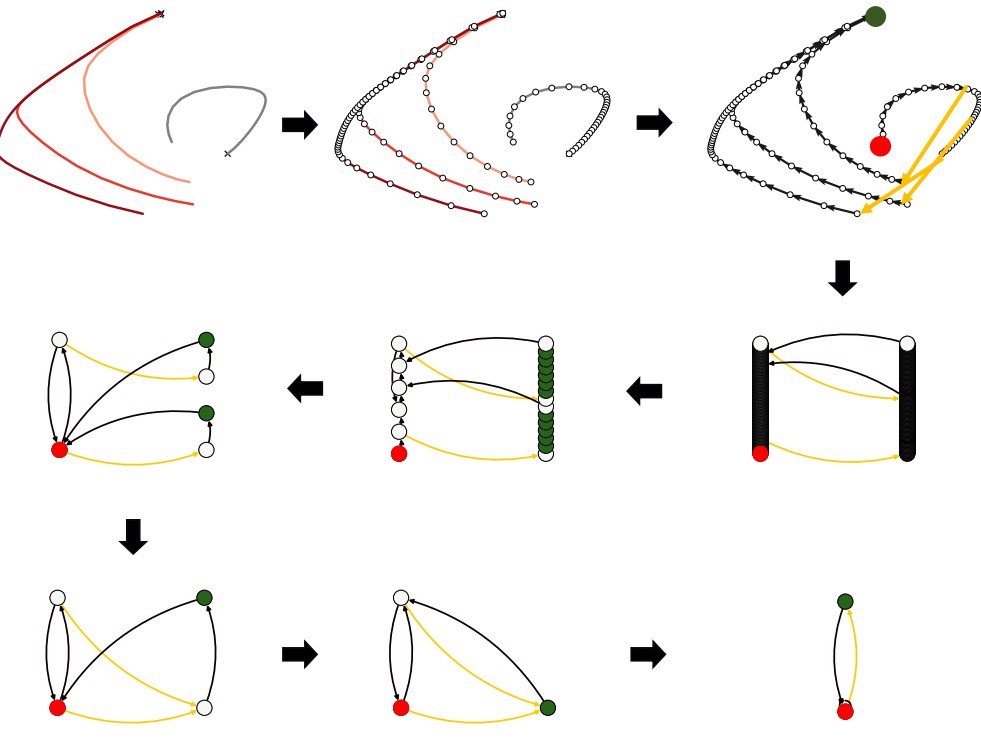

Figure 7: Stages of reduced-dynamics applied to an example network

each trajectory within the manifold changes as a function of the task parameter ($p$) it corresponds to, by fitting a linear regressor to the mapping $p \rightarrow ||\text{Manifold}(p)||_2^2$ and extracting its slope as a feature. For each trajectory within the manifold, we left out the first and last five time steps.

**Cross epoch features**  Here, we focused on the relationship between the trajectory of the one-dimensional epoch and the longest trajectory out of the 2D epoch. We extracted as features the Pearson correlation and the angle between them, the ratio between their speeds, and the margin of their separating hyper-plane obtained from Linear SVM.

## 2.4 Different views of same object

To see whether the neural data from the training set contains information about the topology of the networks, we evaluated the ability of the neural features to predict the reduced dynamics we described earlier. This was done by a cross-validation procedure that included 50 repetitions of fitting a Random-Forest classifier to a randomly selected $90\%$ training set of the networks. We evaluate the *Kappa-Cohen* score (15) and the confusion matrix of the classification on the remaining $10\%$. The results of averaging across all repetitions appear on Figures 8 to 11 and in the main text.

## 3 Variability in Context-dependent integration

Not all tasks display qualitative variability (14). Even without such variability, there can be substantial quantitative variability. Here we highlight several features of such variability (Figure 12).

We train several independently initialized vanilla (11) networks on the context-dependent integration (CDI) task, following the task protocol from (14). All networks form an approximate line attractor (14). For each network, we inject zero channel input along with one of the context, and thus allow

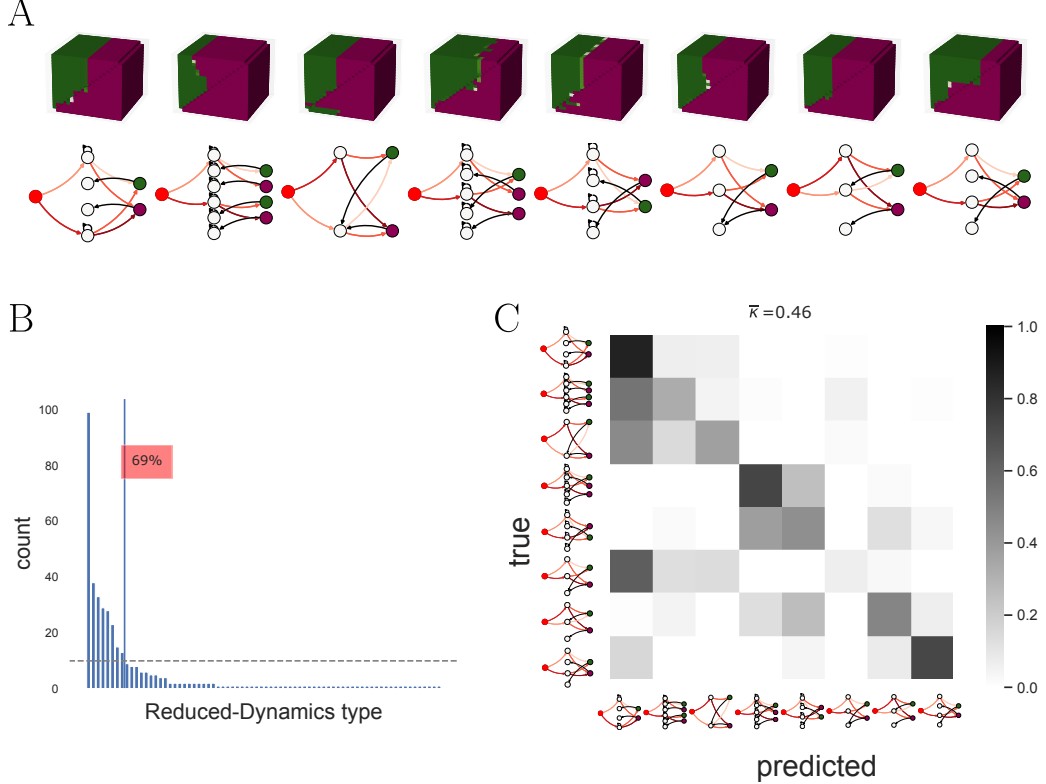

Figure 8: The space of solutions for the delayed discrimination task. **A** Representative extrapolation plots (top) and reduced dynamics graphs (bottom) for the eight most common solution types. **B** Distribution of solution types for the 400 networks trained. The eight solutions shown account for 69% of the networks. **C** Neural features obtained during the training set can partially predict the solution type that includes extrapolation dynamics. The confusion matrix shows the result of this prediction.

the network to converge to the origin of the line attractor (13). Following previous works, we analyze the linearized system around this point. We characterize the network with the following neural measurements.

1. Participation ratio (PR) at t=3: to the linearized system, we deliver an input $u_t \sim \mathcal{N}(0, 1)$ for t consecutive time steps and compute the participation-ratio, a measure of linear dimensionality defined as $\frac{(\sum_i \lambda_i)^2}{\sum_i \lambda_i^2}$ where the $\lambda_i$s are the eigenvalues of $C_t = < h_t h_t^T >_{\text{trials}}$ of the network hidden-state at time $t$ across several trials.

2. Decoder MSE k=3: to the linearized system, we deliver $u_t \sim \mathcal{N}(0, 1)$ for $T = 30$ time steps and we perform linear regression to decode $u_{T-k}$ from $h_T$. We use the decoder MSE as a proxy measure for information held by the network about previous inputs.

3. $\|l_0\|$: norm of the left eigenvector of the linearized system corresponding to eigenvalue with the largest absolute value, i.e the selection vector (16).

4. $\rho(w_{in}^{(0)}, r_0)$: Correlation of input weight vector $w_{in}$ with the right eigenvector of the linearized system corresponding to eigenvalue with the largest absolute value, i.e. the direction along the line attractor (16).

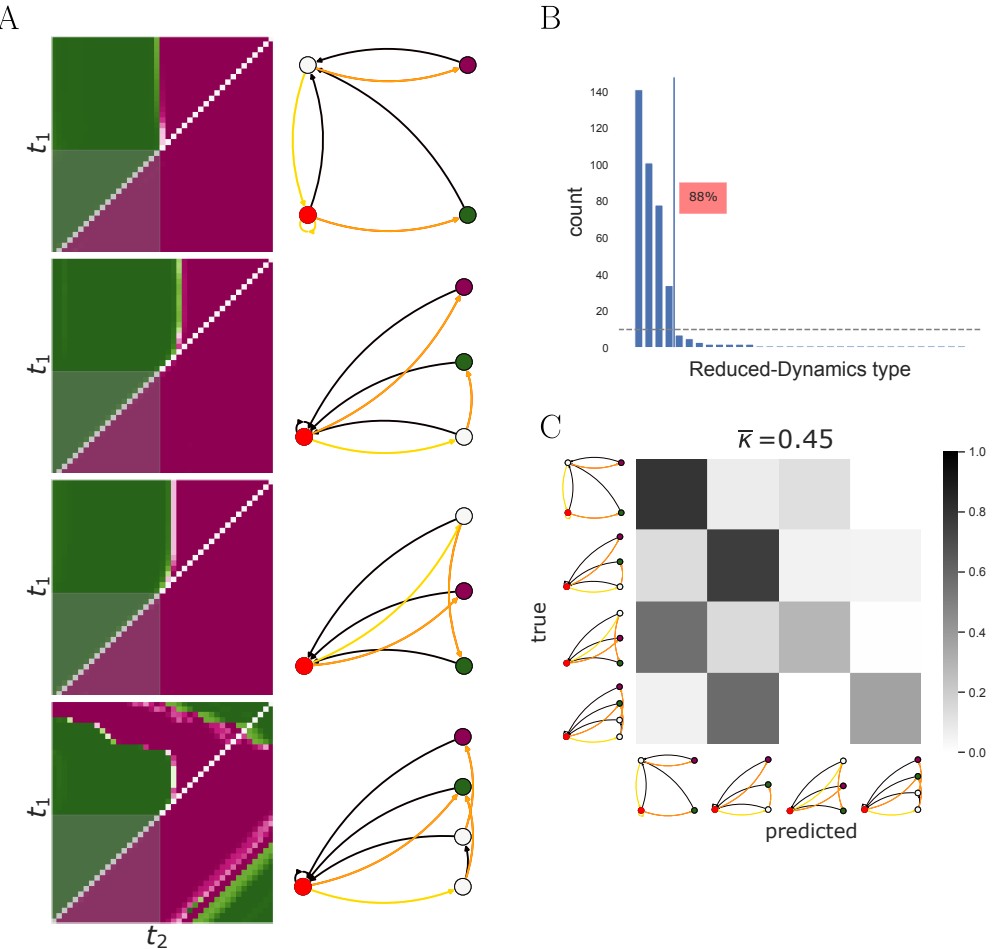

Figure 9: The space of solutions for the interval discrimination task. **A** Representative extrapolation plots (left) and reduced dynamics graphs (right) for the four most common solution types. **B** Distribution of solution types for the 400 networks trained. The four solutions shown account for 88% of the networks. **C** Neural features obtained during the training set can partially predict the solution type that includes extrapolation dynamics. According to the confusion matrix, the classifier is able to discriminate solutions that contain limit cycles from the ones which do not.

5. Second largest $|\lambda_i|$: Eigenvalue of linearized recurrent dynamics with second largest absolute value.

Figure 12 shows each of these neural measures compared to each other.

In addition we evaluate several behavioral measures. We run the networks on a selection of designed inputs. In all cases, we provide input only along one of the channels along with the corresponding context input. The five choices of channel inputs are shown in the top row of Figure 13 (A,B,C,D,E). The inputs are designed such that integral is zero at the end, to facilitate visualization of the error In each case we evaluate MSE of the output from the target of zero. We then compare this MSE with training MSE and the aforementioned neural measurements for each of the networks.

Our neural and behavioral measures indicate a large degree of quantitative variability across networks. The measurements are only weakly correlated to each other, showing that there are many axes of variability. We note that even for networks with very low training MSE, there is still considerable variability in the behavioral challenges (the horizontal nature of the second row of Figure 13). We note, however, that extensive training results in step-like decreases in the training MSE. These steps

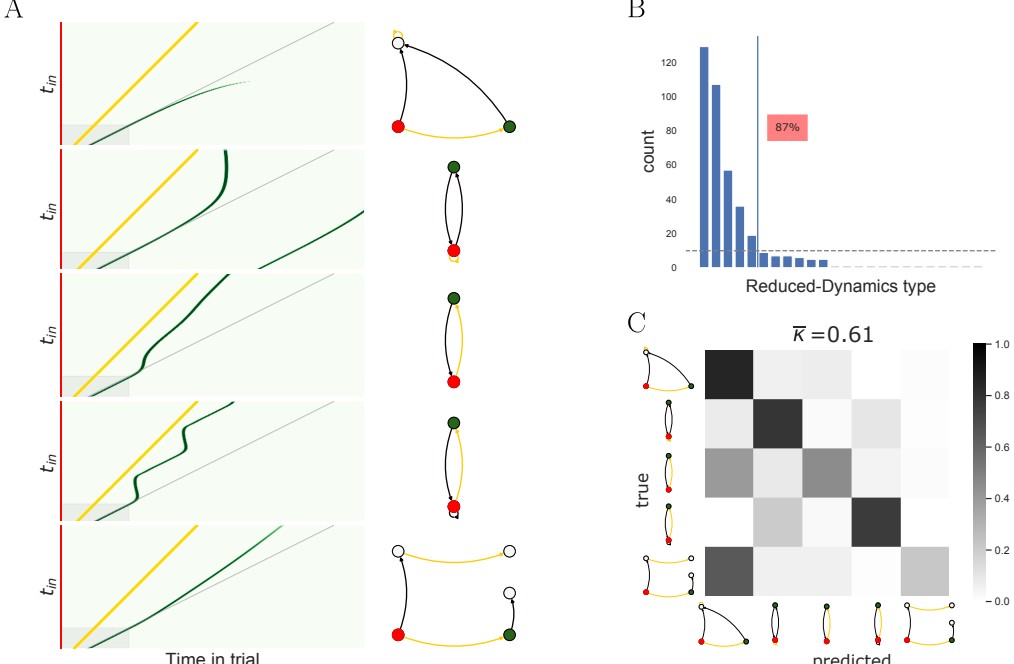

Figure 10: The space of solutions for the time reproduction task, for GRU networks. **A** Representative extrapolation plots (left) and reduced dynamics graphs (right) for the five most common solution types. **B** Distribution of solution types for the 400 networks trained. The five solutions shown account for 87% of the networks. **C** Neural features obtained during the training set can partially predict the solution type that includes extrapolation dynamics. The confusion matrix shows the result of this prediction.

are accompanied by a reduction in the variability of the behavioral challenges, but the latter variability remains much larger for all cases.

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

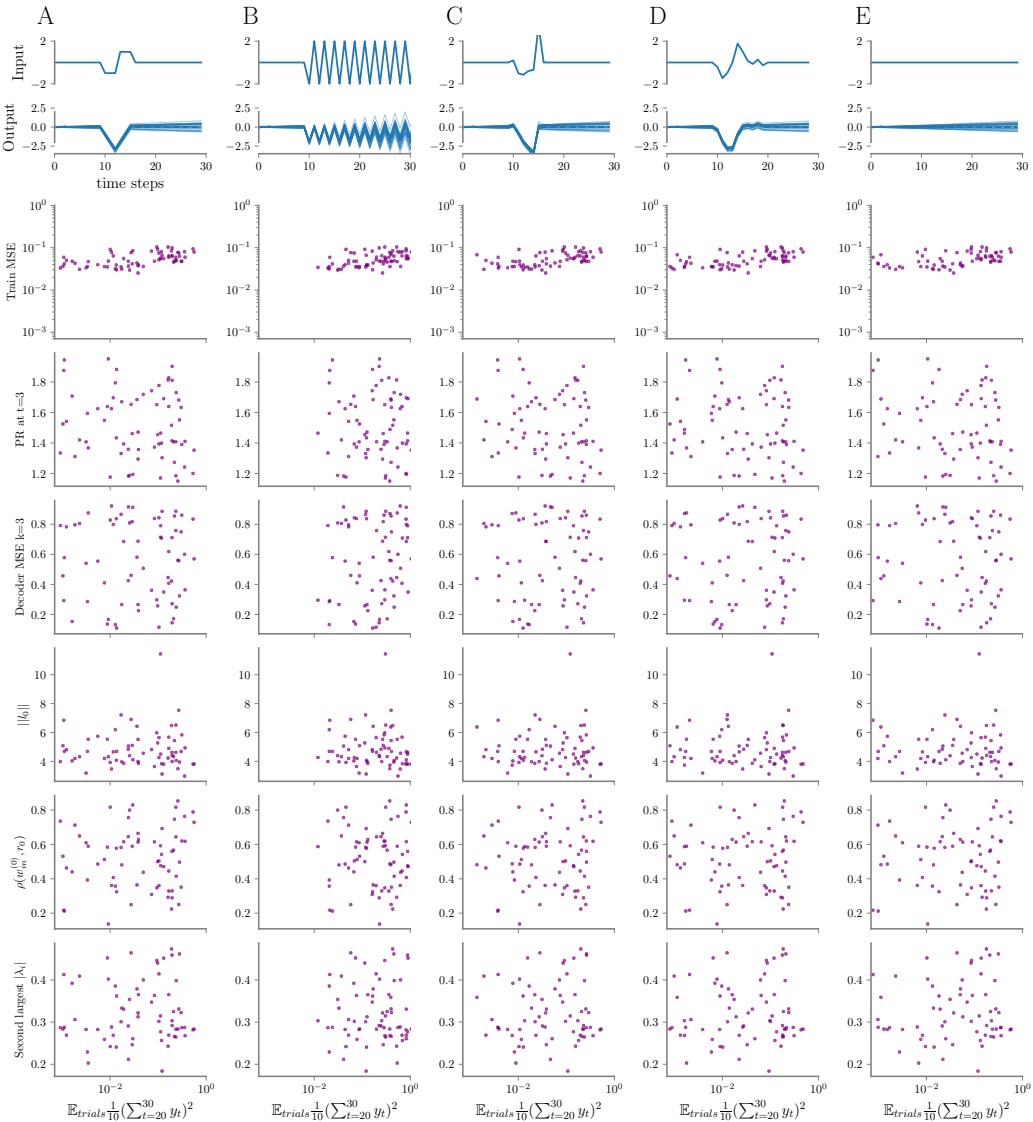

Figure 13: Comparison of neural and behavioral measurements for RNNs trained on CDI (see Figure 12 for details about the colors). We run the networks on five different input protocols, one in each of the columns A,B,C,D,E. First two rows show example inputs and outputs/targets for each of the tasks. Rows 3 to 8 show behavioral MSE for each task versus the neural measurements from Figure 12 wherein each dot indicates measurements for a single network.