# OpenReview forum: "Charting and Navigating the Space of Solutions for Recurrent Neural Networks"
_NeurIPS.cc/2021/Conference — NeurIPS 2021 Poster_

### Official Review · Reviewer_Q7WG · 2021-07-14

**Rating:** 6
**Confidence:** 4

**Summary:**

This study proposes an approach for characterising the space of solutions obtained by fitting neural networks to a couple of tasks (one simple but general task and one neuroscience-related task). The analyses suggest that network initialisation and architecture are two key factors for the degeneracy of solutions obtained when fitting neural networks.

**Limitations And Societal Impact:**

The authors have adequately addressed the limitations of their work.

**Main Review:**

### Originality

The novelty of the study resides in the development of a set of analyses to characterise the space of neural network solutions. Previous work is appropriately cited and it is clear how this work differs from previous contributions.

### Quality

The paper is technically sound, and claims are appropriately backed by empirical evaluation.


### Clarity

The paper is clearly written, and provides enough information for an expert reader to understand all the steps to reproduce the results. There is just a minor typo I would like to point out: in lines 128-134, the authors reference the networks as A to D, and I believe these references to be wrong.


### Significance

I believe the developed analyses would be of interest to the neuroscience community, in particular applied to real neuroscience datasets. However, the conclusions drawn from the analyses are somewhat unsurprising from the Machine Learning point of view, as it is well known that neural network fitting is a problem with multiple local minima, and that this depends on the initialisation and architecture. In fact, it is common and good practice in Machine Learning to train networks several times, with multiple seeds, initialisations and hyperparameters, in order to assess the robustness of the drawn conclusions. In order to improve the significance and impact of the manuscript, I would recommend the authors to apply their set of analyses to a Machine Learning task, with the hope of getting new insights e.g. in terms of optimisation (note that the manuscript is one page shorter than the limit, and thus would have the space for extra analyses).


### After rebuttal

Thank you for the careful answer to all reviews. After reading all the reviews and authors' rebuttal, I believe the authors made an effort to address all the reviews, and in my opinion, to some satisfaction. I increased my score to better reflect the quality and significance of the work.

**Time Spent Reviewing:**

2

---

> ### Author Response · Authors · 2021-08-10
> **Reply to reviewer Q7WG**
>
> We thank the reviewer for the time and comments.
> First, the main changes since the original submission:
> * The 2D example was extended to show conditions without qualitatively different solutions, and analysis showing sources for variable solutions.
> * Two more tasks were studied: An interval comparison task and a sentiment analysis task. Our tools and main findings transferred to these tasks, and we discovered more variability in the solutions of sentiment analysis than was previously reported.
> * The manual topology classification method was replaced by an automatic algorithm operating on well-defined principles.
> Prediction of topologies from dynamical features was done in a more principled and task-agnostic manner.
>  Below we relate to the specific points raised by the reviewer:
>
> We are aware that underspecification is not a novel phenomenon, but would like to stress two points.
> 1. **The implications for neuroscience.** Because previous work focused on tasks for which most solutions were qualitatively the same, trained networks were mostly used to obtain hypotheses on how tasks should be solved by a typical animal/human, instead of harnessing underspecification as a tool to study individual heterogeneity.
> 1. **Machine learning.**  Most work is done on feedforward networks, and we show several properties that are unique to underspecification in recurrent networks. First, the 2D example has an added level of degeneracy for recurrent networks (see response #2 for reviewer H1DK). This is because the same final state can be obtained by different trajectories of the dynamical system. Second, the distinction between solutions in feed-forward networks is typically done only numerically by using generalization error on various test-sets. In contrast, we analyze the mechanisms of the solutions and claim that dynamical objects and bifurcations are natural category boundaries in the recurrent case. Furthermore, the use of extrapolation is a link between these two concepts, as extrapolation is a particular (and in our view well-motivated) example of a test-set.

---

### Official Review · Reviewer_vfv5 · 2021-07-16

**Rating:** 6
**Confidence:** 3

**Summary:**

This paper makes two main contributions. First, the authors study a couple of examples to show that the learned structure and dynamics of an RNN are not completely determined by the task on which it was trained -- legit solutions form a vast space, and where an RNN lands depends on where it starts. Second, on the same examples, the authors set out to dissect how the different solutions differ, both in the algorithmic strategies they employ, and in the realisation of these strategies in the network's internal dynamics. The most novel result is that aspects of the former can be predicted by features of the latter. The first point about degeneracy of solutions is not exactly news, but it is very nicely illustrated here on examples, and it is an important reminder to the field that one should be careful in interpreting the emerging dynamics of trained RNNs as brain-like solutions. The second is potentially of greater practical importance to the field, and I commend the authors for even trying to attack such a difficult problem of relating dynamical motifs at the neural level to "algorithmic strategies"! However, I have many issues with the clarity of the paper, and struggle to understand if the methodology developed in this paper will be of any use beyond the particular Ready-set-go task considered.

**Limitations And Societal Impact:**

Societal impact is not addressed at all, but I think we can live with that given the paper's main topic.
The limitations are discussed.

**Main Review:**

I have found the paper difficult to understand:

- the abstract is cryptic, with intriguing sentences such as "[...] by testing the network's ability to extrapolate, as a perturbation to a system often reveals [...]" → not sure a priori what the link might be between extrapolating and responding to a perturbation; similarly, "we relate extrapolation patterns to specific dynamical objects" → too vague to be useful; similarly, "Using a set of features, we define a space of solutions." → again, too vague to be a useful summary.

- In section 4.3, the reader has to work out for themselves that panels D, E, F and G of Figure 2 refer to networks A, B, C, and D in the main text.

- Figure 3: what are we looking at?? What is the meaning of these circles and squared, open vs filled, coloured vs white? The corresponding Section 4.4 is very opaque -- there are lengthy descriptions of the various possibilities, but no sense of how they were arrived at. Is there a general approach to categorising strategies? "shows the salient locations in phase space" - what does that mean? "enter a slow area" -- where do we see this? "some networks emit a go pulse" -- isn't it the case that all successfully trained networks will emit a go pulse? At this point I am rather lost.

- Figure 5: what's on the x and y axes?

------

Minor:

- Figure 2 : t_s corresponding to each shade of blue in Fig2 D-G should be labelled
- Figure 2H seems a bit arbitrary; why not compute the participation ratio or something?
- Section 4.4: "there are no guarantees" -- for what?


**Time Spent Reviewing:**

4

---

> ### Author Response · Authors · 2021-08-10
> **Reply to reviewer vfv5**
>
> We thank the reviewer for the time and comments.
> First, the main changes since the original submission:
> * The 2D example was extended to show conditions without qualitatively different solutions, and analysis showing sources for variable solutions.
> * Two more tasks were studied: An interval comparison task and a sentiment analysis task. Our tools and main findings transferred to these tasks, and we discovered more variability in the solutions of sentiment analysis than was previously reported.
> * The manual topology classification method was replaced by an automatic algorithm operating on well-defined principles.
> Prediction of topologies from dynamical features was done in a more principled and task-agnostic manner.
>
>
> Below we relate to the specific points raised by the reviewer:
> 1. **Clarity.** We apologize for the lack of clarity and will improve it in the final version.
> 2. **Extrapolation as a perturbation.** The idea is similar to point #1 in the response to Reviewer teJ3. Because all networks perform the task well by definition, the differences in their underlying algorithms can either be revealed by looking at internal (neural) activity, or by challenging them with novel stimuli (extrapolation/perturbation).
> 3. **Topologies and figure 3.** As detailed in the response to reviewer H1DK, we now use a principled algorithm to define topologies and will describe them more clearly. Specifically for the old Figure 3 - all networks emit a Go pulse. But some will do this even without a Set pulse (A, C, F). Some will emit several Go pulses in a limit cycle (A).
> 4. **Figure 5.** We now improved our prediction from dynamics to topology (see point #3 in response to reviewer H1DK). The old figure showed the actual vs. the predicted extrapolation score, where the prediction was done using neural features obtained during training (no extrapolation trials). A perfect prediction would correspond to a diagonal line. The figure shows that we are very far from a perfect prediction (which is reasonable, given that we observe the network in a very different condition), and yet are also very far from a random prediction (Spearman values).
> 5. **Participation ratio in Figure 2H.** Thank you for the suggestion.
> 6. **Other typos.** Will be fixed.

---

> > ### Comment · Reviewer_vfv5 · 2021-08-31
> > **Thank you**
> >
> > Many thanks for your response. I agree with the other reviewers that the new additions to the paper are moving it in the right direction. Nevertheless, I feel I do not have enough information to be able to fully assess the new automated topology classification (which in my mind would be central to the methodology developed in this paper, if it is to be of general use), therefore I am keeping my score at 6.

---

### Official Review · Reviewer_teJ3 · 2021-07-17

**Rating:** 6
**Confidence:** 4

**Summary:**

The authors study the solutions  RNNs find and characterize them based on extrapolation and recurrent dynamics. They primarily consider two cases -- a simple two neuron network performing a memory task, and larger networks performing a Ready-Set-Go task. They show that the final solution depends on the initial connectivity, but not so much on the hyperparameters of learning.


**Limitations And Societal Impact:**

The authors have a fairly good discussion of the limitations of the paper.

**Main Review:**

**Update:** Based on the authors response, and new results mentioned, I have updated my score to reflect my more positive evaluation of the paper.

Overall, this is very interesting and nicely done work but the paper has some flaws that resulted in my recommendation of rejection.

## Originality:
There are other works characterizing RNN dynamics, e.g. (Maheswaranathan et al. 2019) which is cited in the related work. All the aspects where this work differs is not very clear (might be useful to add), but one original contribution of this paper seems to be the use of extrapolation for characterizing the solution, which is an interesting idea. Another contribution is to show that initial connectivity and not hyperparameters influence the final solution (although it remains to be seen how generally applicable this is).

## Quality:
The experiments performed in the paper are well defined, and the conclusions reached are reasonable. But the take-home message of the paper is not clear. Concrete comparisons with biological data would have strengthened the paper quite a lot.

One specific comment: It's strange that the parameters are trained in the PC space for the 2-neuron network as described in the caption for Fig. 1. Training in the original weight space and projecting onto the PC space just for plotting would have been more meaningful.

## Clarity:
The writing is reasonably clear, but there are some gaps in the explanations:
* How are the classes of networks defined? Is there a quantitative measure for this? Are the labels done manually? This key detail is not very clearly spelled out.
* Sec. 4.3: Details about the networks being tested are not given. How large are these networks, and of what type?
* Fig. 5 has no axis labels and explanation of what's plotted
* l.219: Not sure reference [18] makes sense in the context of that sentence.

Minor:
* Fig. 2A: $t_p$ should be $t_s$?
* Lines 128-135: Networks are referred using A,B,C,D while the figures are labelled D-G causing some confusion.
* l.152: Figure 2A should be Figure 2D.

## Significance:
Understanding the space of solutions of RNNs is a very important problem, and this paper adds a couple of insights related to this.
Another key contribution of the paper is to show the variety of solutions that emerge for the considered tasks -- highlighting the selection bias present in papers that show activity similar to biology emerging in networks trained to do specific tasks.

**Time Spent Reviewing:**

5

---

> ### Author Response · Authors · 2021-08-10
> **Reply to reviewer teJ3**
>
> We thank the reviewer for the time and comments.
> First, the main changes since the original submission:
> * The 2D example was extended to show conditions without qualitatively different solutions, and analysis showing sources for variable solutions.
> * Two more tasks were studied: An interval comparison task and a sentiment analysis task. Our tools and main findings transferred to these tasks, and we discovered more variability in the solutions of sentiment analysis than was previously reported.
> * The manual topology classification method was replaced by an automatic algorithm operating on well-defined principles.
> Prediction of topologies from dynamical features was done in a more principled and task-agnostic manner.
>
> Below we relate to the specific points raised by the reviewer:
> 1. **Distinctions from Maheswaranathan et al 2019.** Briefly, we study the same problem and find opposite results. We believe this is due to two reasons. First, the tasks used in that study have stronger constraints on their dynamics. It is very hard to imagine a different dynamical structure for solving the flip-flop task. It might be that the space of solutions for these tasks is in some sense convex - at least regarding qualitatively different solutions. Our focus on a timing task revealed cases where this is not the case. The 2D example shows how such variability arises, and we now also show a 2D case, more similar to the flip-flop, in which there is no qualitative variability. A second possible reason is how we probe the networks. Extrapolation, or more generally unexpected stimuli, can reveal underlying variability that is harder to detect otherwise. We now train a sentiment analysis task, similar to that of Maheswaranathan et al, and show that probing it with custom stimuli can reveal hysteresis, indicating that the line attractor has some thickness, and different histories can lead to different locations in this other dimension. Crucially, the thickness profile varies between networks. While not a strong qualitative difference as those observed in the timing task, we believe this indicates the utility of custom stimuli to reveal underlying variabilities.
>
> 2. **Initial connectivity vs. hyperparameters.**  We did not claim that hyperparameters do not matter, but rather that even if they are held fixed, initial connectivity can still lead to substantial variability. The generality of this statement is now strengthened by two additional tasks - the sentiment task described above, and an interval discrimination task that is described in the answer to reviewer H1DK.
> 3. **Relating to experimental data.** Great suggestion. Beyond the scope of the current work. We believe this lays the foundation for such work. Especially because existing studies mostly focus on averages, rather than on heterogeneity (see review by Musall et al, curr. Opinion Neurobiology,  2019)
> 4. **Training in PC space.** We apologize for the lack of clarity. The training was done in the full space, and PCA was used for visualizations. In one instance, we also used PCA to choose initial conditions, and then performed gradient descent on all parameters.
> 5. **Definition of classes.** This was done semi-manually. Now done in a principled and automatic manner, as described in the answer to reviewer H1DK.
> 6. **Network details.** For each combination of architecture (Vanilla, GRU, LSTM) and a hidden units size (20,30,40,50) we trained 100 networks (1200 in total). The exact details are in the supplementary material, section 2.1.
> 7. **Other typos.** Will be fixed. Thanks for noticing.

---

> > ### Comment · Reviewer_teJ3 · 2021-08-27
> > **Most concerns addressed**
> >
> > Thank you for your response. Considering the significant additional work done in adding new tasks (including and especially the delayed discrimination task), automating the definition of classes and addressing other concerns, I am willing to upgrade my score to an accept.

---

### Official Review · Reviewer_H1DK · 2021-07-22

**Rating:** 5
**Confidence:** 5

**Summary:**

This paper analyzes the topology of the solutions of RNNs trained on time counting tasks, revealing a rich set of possible dynamical solutions and architectural biases that affect the generalization properties of the solution. It uses numerical experiments to illustrate the diversity of solutions that can arise for tasks where the solution relies on slow transients rather than fixed points for performing the task and highlights some of the difficulties in performing a systematic analysis in such contexts. This is potentially relevant for understanding across subject variability in animals performing such tasks.

**Ethical Concerns:**

No  ethical concerns.

**Limitations And Societal Impact:**

No obvious negative social impact. The limitations of the approach are touched upon in the discussion but not very satisfactorily.

**Main Review:**

Overall evaluation: An interesting idea but the lack of systematicity in the approach diminishes the value and interpretability of the results. More work needed before publication.

Major concerns: While liking the general idea, there's not a sense that we have really understood the phenomenology by the end of the paper.  The analysis feels incomplete, and the described topological structure somewhat anecdotal. I would have thought that the 2-neuron network toy example would have provided more analytical backing for the arguments but is seems to also primarily rely on simulations to make the points. Are the topological features in fig3 hand designed by the scientist as a human interpretation of continual trajectories or could this topological structure be identified algorithmically? For the last part when identifying dynamic signatures of the solution that can be used to differentiate various algorithmic solutions (which are defined 'manually' by the scientist based on the same trajectories), I am unsure what have we actually learned there. Is there a way to define classes of solutions straight from data and to which degree are these methods applicable to neural data (i.e. are these quantities that one could reasonably estimate from limited neural recordings?). Also not sure what the take home message is for the behavioral predictions in fig 6.

Significance: highlights the computational distinctions across 'working memory'-like tasks and furthers the arguments for a topological/algorithmic understanding of RNNs computations. Missed opportunities in relating to across animal variability in similar neuroscience experiments.

Originality: Since the number of degrees of freedom in trained RNNs is often much larger than the task constraints, there is an increasing interest in characterizing the nature of the solutions trained by RNNs and their generalization properties, with relevance for both neuroscience and machine learning. In particular, Susillo et al have shown that certain memory task show a certain form of universality, where the qualitative nature of the solution is more or less the same regardless choices of architecture and hyperparameters. This paper uses similar ideas to show that other tasks that count time are less constrained in the nature of their solution with several distinctive algorithms.

Clarity: The abstract and introduction are very well written, but the related literature lacks substance and the discussion is poorly structured and needs a fair bit of work.  The quality of the figures is poor, with systematically missing axis labels, inconsistent colors across panels, etc; it would also greatly help to have in figure legends for different colors. the captions help somewhat to make sense of the lot but makes the reader's life unnecessarily difficult.

**Time Spent Reviewing:**

3

---

> ### Author Response · Authors · 2021-08-10
> **Reply to reviewer H1DK**
>
> We thank the reviewer for the time and comments.
> First, the main changes since the original submission:
> * The 2D example was extended to show conditions without qualitatively different solutions, and analysis showing sources for variable solutions.
> * Two more tasks were studied: An interval comparison task and a sentiment analysis task. Our tools and main findings transferred to these tasks, and we discovered more variability in the solutions of sentiment analysis than was previously reported.
> * The manual topology classification method was replaced by an automatic algorithm operating on well-defined principles.
> Prediction of topologies from dynamical features was done in a more principled and task-agnostic manner.
>
> Below we relate to the specific points raised by the reviewer:
> 1. **The topological structure is anecdotal, hand-designed.** We agree and therefore designed a principled algorithm that operates in two stages. First, converging trajectories from different trials are identified, and a graph of the phase space of the system is constructed, in which nodes are states and edges define transitions due to time or stimulus. At the second stage, the graph is systematically reduced by collapsing parallel paths along the graph. In this manner, only qualitative differences remain. The result is very similar to the manual classification appearing in the current version but does not require any hand-tuning of features or classes.
> 2. **The 2 neuron example.** The example illustrates that very simple tasks can already generate qualitatively different solutions. We added an analysis of this degeneracy for both feed-forward and recurrent networks. Namely, the task is
> $\begin{bmatrix}0 \\\\ 1\end{bmatrix} = A \begin{bmatrix}1 \\\\ 0\end{bmatrix}$, where the matrix $A\in \mathbb{R}^{2\times2}$ is either the parameter matrix (feed-forward), or $A=W^n$ for a recurrent network that propagates n steps in time. The former case has two unconstrained parameters ($A_{12}$, $A_{22}$), and therefore a 2D space of solutions. The latter case has two n^th order equations and therefore can generate an additional level of redundancy in solutions. This is the underlying reason for the limit cycle solutions, where the different frequencies correspond to different roots of these equations.
> 3. **Dynamic signatures.** The aim here was to show that experimentally accessible measurements, taken during the normal training regime, can predict the topological class that is defined via dynamics outside this regime (in the extrapolation regime). We changed our previous definition of neural features to one that is less hand-tuned and that can generalize to other tasks. Specifically, we only assume the task can be divided into two epochs, and all the measures we use relate to geometrical features of the activity during these epochs and relations between them. This renders the measures agnostic to the details of the task and generalizable. As further proof of the generality of these measures, we also used them on a new task (interval discrimination) and were able to predict topological classes without fine-tuning either the dynamic observables or the topological algorithm.
> 4. **Relating to experimental data**. This is a great suggestion, but we feel it is beyond the scope of the current work. We believe our work lays the foundation for such work. Especially because existing studies mostly focus on average behavior rather than harnessing heterogeneity between individuals (see for instance the review by Musall et al, cur opinion in neurobiology 2019).
> 5. **Clarity.** We apologize for the quality of figures and the lack of adequate literature review - these will be fixed.

---

> > ### Comment · Reviewer_H1DK · 2021-08-11
> > **Post-rebuttal comments**
> >
> > Thank you for the feedback.
> > I welcome the transition to algorithmic feature identification, which significantly strengthens the approach.
> > Less convinced that the addition of sentiment analysis makes sense in the task mix.
> > Still think not making the link to experimental data in the context of working memory tasks in the discussion is a missed opportunity.
> > Overall, the author replies reassure me in some respects, but given the amount of work required to get the manuscript into shape this remains just below the bar, even with the promised corrections. I will change my overall score to a 5 to reflect this.

---

### Author Response · Authors · 2021-08-26
**Results from another task analyzed**

Dear Reviewers,

We are aware that the discussion period is near its end, but would like to bring to your attention the results of an additional task we analyzed.

The main tasks analyzed thus far involved timing, and we wanted to verify that our observations also hold for tasks in which timing plays no role, and the natural solution involves a line attractor.

To this end, **we trained networks on a delayed discrimination task** (Romo et al. Nature, 1999). The task consists of two input pulses of varying amplitude separated by a delay, and the desired output is an indicator of whether the first stimulus is larger than the second one. We used **variable delays** to discourage networks from reaching an oscillatory solution in which the desired memory is only available at a specific time interval (Orhan and Ma, Nature Neuroscience, 2019). We found that networks converge to **at least two qualitatively different solutions**. These differences were evident in the network dynamics during the original training stimuli, without extrapolation. One solution involved a **line attractor**, where each slow point corresponds to a different value of the first stimulus. Another solution that emerged was a **line of limit cycles**, where each limit cycle corresponds to a different stimulus value, and the oscillation is roughly orthogonal to the direction of the line. Similar to the examples presented in our paper, **we could classify these solutions according to dynamical signatures in the training set**. Furthermore, **we could design test-stimuli that reveal the nature of the solution**. Specifically, injecting noise during the delay had an oscillatory effect on classification performance in one type of solution.

We believe this example strengthens our main message - canonical neuroscience tasks admit multiple, qualitatively different solutions. These solutions can be classified by examining network dynamics, and can also be revealed behaviorally by designing perturbative input trials.

We hope these new results strengthen the generality of the paper, and in any case thank you for your time and effort in reviewing it.

---

> ### Comment · Reviewer_H1DK · 2021-08-26
> **new task**
>
> This sounds like a much more interesting task to look at; and the results are intriguing. definitely on the right track with this.

---

### Decision · Program_Chairs · 2021-09-28

**Decision:**

Accept (Poster)

**Comment:**

The authors present in this work an analysis of the multiple solutions that can be identified by a recurrent neural network trained on a task. in particular, the authors discuss the implications for computational neuroscience studies that leverage such tools. The reviewers all agree that the work was well presented, and in particular strengthened by a second analysis performed in response to the reviewer comments that remove a potential confound of timing. While the scores themselves seem to be borderline, the topic is very timely and the results appear to be highly relevant to the use of ML in Neuroscience. I therefore recommend this work (with the addition of the extra analysis) be accepted to NeurIPS.

**Consistency Experiment:**

NeurIPS has a long history of experimentation. In 2014, NeurIPS ran an experiment in which 10% of submissions were reviewed by two independent committees to quantify the randomness in the review process. This year, we repeated a variant of this experiment to see how the quality of the review process has changed over time.  This paper was part of the experiment and was therefore assigned to two committees (consisting of reviewers, an Area Chair, and a Senior Area Chair) that reached independent decisions.  If both committees made the same recommendation, this recommendation was followed. If a single committee recommended acceptance, the paper was accepted (with the exception of a few cases in which the other committee identified what we considered a fatal flaw, e.g., an error in a key result).

This copy’s committee reached the following decision: **Accept (Poster)**

The other committee assigned to the paper recommended **Reject**.  You can find the other set of reviews, along with any follow up discussion with the authors here:
https://openreview.net/forum?id=SQm_poGrlj